# What Is the Comparative Efficacy of Surgical, Endoscopic, Transanal Resection, and Radiotherapy Modalities in the Treatment of Rectal Cancer?

**DOI:** 10.3390/healthcare11162347

**Published:** 2023-08-20

**Authors:** Alexandru Isaic, Alexandru Cătălin Motofelea, Dan Costachescu, Gheorghe Nicusor Pop, Bogdan Totolici, Dorel Popovici, Razvan Gheorghe Diaconescu

**Affiliations:** 1IInd Surgery Clinic, Timisoara Emergency County Hospital, 300723 Timisoara, Romania; isaic.alexandru@umft.ro; 2Department X of General Surgery, “Victor Babes” University of Medicine and Pharmacy, 300041 Timisoara, Romania; 3Department of Internal Medicine, Faculty of Medicine, “Victor Babes” University of Medicine and Pharmacy, 300041 Timisoara, Romania; 4Department of Orthopedics-Traumatology, Urology, Radiology, and Medical Imaging, “Victor Babes” University of Medicine and Pharmacy Timisoara, Eftimie Murgu Square 2, 300041 Timisoara, Romania; costachescu.dan@umft.ro; 5Department of Oncology, “Victor Babes” University of Medicine and Pharmacy, 300041 Timisoara, Romania; dorel.popovici@umft.ro; 6Center for Modeling Biological Systems and Data Analysis, “Victor Babes” University of Medicine and Pharmacy, 300041 Timisoara, Romania; pop.nicusor@umft.ro; 71st Clinic of General Surgery, Arad County Emergency Clinical Hospital, 310158 Arad, Romania; totolici_bogdan@yahoo.com; 8Department of General Surgery, Faculty of Medicine, “Victor Babes” Western University of Arad, 310025 Arad, Romania; 9OncoHelp Hospital, 300239 Timisoara, Romania; diaconescu.razvan-gheorghe@student.uvvg.ro; 10Department of Surgery, Faculty of Medicine, “Victor Babes” Western University of Arad, 310025 Arad, Romania

**Keywords:** TME, TaTME, TEM, radiotherapy

## Abstract

Background: Rectal cancer is a significant healthcare burden, and effective treatment is crucial. This research aims to compare the effectiveness of surgical and endoscopic resection, transanal resection, and radiotherapy. Methods: A literature analysis was conducted in order to identify relevant studies, by comparing the different surgical approaches and variables affecting treatment decisions. The findings were analyzed and synthesized to provide a comprehensive overview. Results: Surgical treatment, particularly TME (total mesorectal excision), proved consistent efficacy in achieving complete tumor resection and improving long-term survival. Endoscopic treatment and transanal resection techniques were promising for early-stage tumors but were associated with higher local recurrence rates. Radiotherapy, especially in combination with chemotherapy, played a crucial role in locally advanced cases, improving local control and reducing recurrence risk. Patient data, tumor characteristics, and healthcare system factors were identified as important factors in treatment modality selection. Conclusion: Surgical treatment, specifically TME, remains the recommended standard approach for rectal cancer, providing excellent oncological outcomes. Endoscopic treatment and transanal resection techniques can be considered for selected early-stage cases, while radiotherapy is beneficial for locally advanced tumors. Treatment decisions should be personalized based on patient and tumor characteristics, considering the available resources and expertise within the healthcare system.

## 1. Introduction

Rectal cancer is a significant global health issue, targeting the rectum and comprising approximately ten percent of all identified cancer cases [1]. Its incidence, complex management, and resource utilization contribute to the substantial healthcare challenge to healthcare systems worldwide. As rectal cancer is becoming more common, it implies the need for effective procedures and sufficient resources to meet the needs of people who are afflicted. Rectal cancer can represent a challenge for existing curative treatment options due to its location within the bony pelvis and proximity to important organs including the uterus, bladder, and prostate [2]. It also often manifests at an advanced stage, making care more difficult and decreasing the likelihood of a favorable course of therapy [3]. Late-stage diagnosis can occur due to various factors, including the absence of early symptoms or delayed medical attention. Consequently, rectal cancer may progress, necessitating more aggressive treatment approaches and potentially affecting patient outcomes. In addition to the impact caused by the disease, rectal cancer significantly affects both the well-being and chances of survival for individuals affected by it. Rectal bleeding, altered bladder habits, stomach discomfort, and unintentional weight loss are only a few symptoms that may seriously affect everyday life and general wellbeing [4]. This further intensifies the challenging experience for individuals undergoing cancer treatment. In the management of rectal cancer, several main treatment modalities are commonly employed. Surgical treatment plays a pivotal role, and the most frequently performed surgeries for rectal cancer include TME [5]. TME is a meticulous surgical technique that involves the complete removal of the rectum along with the surrounding lymph nodes, ensuring clear margins and minimizing the risk of local recurrence. Ultimately, radiation plays a crucial role in the management of rectal cancer, frequently employed alongside surgical intervention to improve local disease management and reduce the likelihood of relapse [6]. In light of these factors, the purpose of this study is to examine the effectiveness of different rectal cancer treatment procedures, such as TME (total mesorectal excision), TaTME (transanal total mesorectal excision), TEM (transanal endoscopic microsurgery), and radiotherapy. By comparing the effectiveness of these strategies, we hope to acquire a better understanding of their strengths and shortcomings in various clinical circumstances. Acquiring this knowledge will bolster the efficacy of treatments for individuals with rectal cancer, empowering evidence-driven decision-making processes and providing valuable support. This evidence-driven approach will empower healthcare professionals to make informed decisions and provide better support to those facing rectal cancer.

## 2. Background 

Surgery, radiation therapy, and chemotherapy are the presently used conventional treatments for rectal cancer, with surgery as the cornerstone of treatment. The recommended standard surgical technique is TME, which consists in the removal of the entire rectum along with the surrounding lymph nodes and fatty tissue to achieve clear margins [7,8]. Another treatment modality is endoscopic treatment, which involves the use of specialized instruments to remove or treat early-stage rectal tumors. Transanal resection is a procedure specifically designed for the management of rectal tumors located close to the anus. Transanal resection techniques are becoming increasingly popular for treating rectal conditions. These include TaTME (transanal total mesorectal excision), TME (total mesorectal excision), and TEM (transanal endoscopic microsurgery). They provide several potential benefits. TaTME provides enhanced visualization and access to the rectal area through the anus, allowing for precise dissection and removal of tumors [9]. It aims to achieve complete mesorectal excision. Additionally, the minimally invasive nature of TaTME reduces surgical trauma, leading to less postoperative pain, reduced hospital stays, and improved recovery [10]. TME improves oncological outcomes and reduces the risk of local recurrence [11]. This approach has demonstrated lower morbidity and mortality rates compared to older surgical techniques, improving long-term survival for rectal cancer patients. TEM offers a minimally invasive approach for the removal of rectal tumors. Using specialized instruments and an endoscope, TEM allows for precise tumor excision through the anus [12]. TEM is commonly employed for early-stage rectal tumors confined to the rectal wall, offering a less invasive alternative to extensive surgeries like TME [12].

Rectal cancer can be influenced by various determinants. It tends to favor patients with earlier-stage diagnoses as opposed to those who present an advanced-stage diagnosis during screening [13]. As per recent advancements within healthcare fields, supporting early-detection measures and introducing more effective procedures, such as treatment methods, has led to a marked increase in survival rates for rectal cancer cases over time. Several factors can impact one’s chances of recovery from rectal cancer. The factors affecting cancer prognosis are varied and intricate. Tumor stage, response to therapy, genetic mutations, and the patient’s overall health all contribute to outcome prediction.

### 2.1. Pathology

Rectal cancer develops in the rectum. The disease occurs when malignant cells grow uncontrollably from the tissue lining the inside of this area [1,2]. Initially, normal cells in the rectal lining undergo mutations, forming polyps that can eventually turn into malignant tumors over time [14]. Key genes responsible for regulating cell division, such as APC, KRAS, and TP53, are often mutated during this process [14]. If left untreated, rectal cancer can progress locally, invading nearby tissues and structures, and can also spread to regional lymph nodes and distant organs through the bloodstream or lymphatic system.

### 2.2. Epidemiology of Rectal Cancer

The incidence of rectal cancer shows significant geographic and demographic disparities. Western countries generally have a higher incidence than Asian countries [15]. Possible factors contributing to this disparity could be the existing differences in diet and lifestyle [15]. The number of new instances of colon cancer is expected to rise up to 106,970 by 2023, with rectal cancer accounting for around 46,050 of those new diagnoses in the United States. This surprising statistic underscores colo-rectal cancer as the third most prevalent cancer worldwide [13], emphasizing its significant impact on global health. Rectal cancer affects 15 to 25 people per 100,000 individuals yearly within the European Union. Unfortunately, a third of those affected pass away due to this disease annually [16]. Rectal cancer often affects people over 70 years, but this threshold is projected to rise. The estimated yearly death rate is 4–10 individuals per a hundred thousand people. However, significant variance persists among five-year survival statistics depending on the distinctiveness observed across underlying conditions. For localized cancer (confined to the rectum), it may be around 90% [13]. But if it spreads to lymph nodes or other organs, these rates decrease significantly for localized rectal cancer. Over the last two decades, stages II–III rectal cancer has had a five-year OS rate of 65% [17]. For stage IV of rectal cancer, the 5-year relative survival rate is approximately 17%, according to the American Cancer Society [18]. Although the prevalence of rectal cancer has decreased due to widespread screening measures facilitating the detection and management of premalignant lesions [19], recent studies have indicated a rise in its occurrence among younger individuals. Furthermore, it is projected that the incidence of rectal cancer specifically will rise by 124.2% for patients aged 20–34 years by 2030 [20]. In regions like Hong Kong, colon and rectal cancers show distinguishing patterns in incidence. This implies that these two cancer forms have different underlying causes [21]. A gap exists among individuals under 50 years due to the rising prevalence of rectal cancer in younger people [6]. In addition, compared to Caucasians, Black people have a higher incidence of rectal cancer [7]. Survival outcomes among younger patients with locally advanced rectal cancer also demonstrate racial disparities favoring the Caucasians [22]. Additionally, rectal cancer incidence shows temporal trends characterized by a general decline but an increase in young adult population countries such as Australia, Canada, and the United States [23].

### 2.3. Risk Factors

Rectal cancer can stem from various risk factors, out of which the advanced age is a significant one. Colorectal cancer risk is significantly associated with dietary habits characterized by excessive consumption of processed foods and red meats, inadequate fiber intake, and a deficiency in fruits and vegetables [24]. A hazard may also come from a sedentary lifestyle or harmful habits like obesity, smoking, or binge drinking [25,26]. Research uncovered a dose–response connection which suggested that moderate alcohol intake reduces the overall risk, while excessive consumption of alcohol raises the risk for colorectal cancer [15].

### 2.4. Diagnosis

The diagnosis involves multiple steps. Initially, methods like barium enema or CT colonography may be employed, but a colonoscopy is ultimately necessary to obtain a tissue sample for biopsy [14]. While flexible sigmoidoscopy can help reduce CRC mortality, it cannot fully replace a complete diagnostic colonoscopy [14]. In cases where colonoscopy is incomplete, capsule endoscopy has been approved by the FDA (Food and Drug Administration) as an alternative [14]. Although routine laboratory tests are useful, they do not provide a definitive diagnosis [14]. CT scans can assess the tumor and lymph node stages and detect distant metastases with good sensitivity [14]. While PET scanning is not commonly used in preoperative staging, a biopsy of suspicious metastatic sites is still necessary to confirm the diagnosis [14].

## 3. Materials and Methods

### 3.1. Literature Search Strategy

To conduct a comprehensive literature search, several databases were used, including PubMed, Google Scholar, Web of Science, NCCN, ESMO, and NCBI. These databases were selected because they offer thorough coverage of the biomedical literature, including peer-reviewed publications and clinical trials. The following search terms were used: “rectal cancer”, “rectal tumor”, “surgical treatment”, “endoscopic treatment”, “transanal resection”, “efficacy”, ”TaTME”, ”transnal total mesorectal resection”, “TME”, “total mesorectal excision”, “TEM”, “total endoscopic microsurgery”, and “outcome”. The search terms were combined using the operators (AND, OR) to expand the search. 

### 3.2. Inclusion and Exclusion Criteria

Studies were included if they met the following criteria: (1) written in English, (2) clinical studies, (3) required human subjects with rectal cancer, (4) compared the efficacy of different treatment modalities (surgical treatment, endoscopic treatment, transanal resection, and radiotherapy), (5) treatment effectiveness results presented, including tumor response, local recurrence, overall survival, and disease-free survival, (6) included patients with Stage I, Stage IIA, Stage IIB, Stage III and Stage IVA, and (7) studies conducted from 2013–2023. Any research that was conducted on animals, presented only in an abstract form, was a case report, editorial, or conference abstract was disregarded. We excluded studies that lacked measurable data on efficacy, survival, recurrence rates, or other significant clinical outcomes. Also excluded were those focused solely on patients with benign or precancerous lesions, and those investigating experimental or novel treatments not currently in standard clinical use. Duplicate studies and studies with insufficient data were also excluded.

### 3.3. Data Extraction

A standardized data extraction form was utilized to ensure consistency in the process. The form included fields for extracting relevant information from the selected studies. Data extraction involved recording patient characteristics (age, cancer stage), treatment methods examined in each study, primary outcomes assessed such as (tumor response rates, survival rates, recurrence rates), and key conclusions and findings from each study. The initial search yielded 5677 articles. After removing the duplicates, case reports, studies with animals, abstract forms, editorials, conference abstracts, those focused solely on patients with benign or precancerous lesions, and those investigating experimental or novel treatments not currently in standard clinical use were excluded. Following a comprehensive search within the scope of our subject, a refined selection of 83 articles was identified for further in-depth review and analysis.

### 3.4. Data Analysis

A descriptive analysis was conducted to summarize the characteristics and outcomes of the included studies. This analysis aimed to provide a comprehensive overview of the findings and key results obtained from the literature. The extracted data were analyzed through a descriptive analysis to summarize the characteristics of the included studies (study design, sample size, patient population), the outcomes and results of the different studies, and any overarching trends, patterns, or themes that emerged across the multiples studies. The focus of the literature review was to synthesize the evidence on treatment efficacy, specifically looking at outcomes such as local recurrence rates for different treatment modalities, overall and disease-free survival rates, tumor response and resection rates, and factors influencing treatment decisions and patient selection. Both early-stage and advanced rectal cancers were considered in the analysis.

## 4. Comparative Analysis 

Surgical treatment and endoscopic treatment represent two distinct modalities in the management of rectal cancer, each with its own set of advantages and considerations. Surgical treatment, particularly TME, is widely regarded as the standard approach due to its ability to achieve complete tumor resection and excision of surrounding tissues, resulting in favorable oncological outcomes. However, complete tumor resection is also achieved by endoscopic treatment and other techniques. 

Comparing surgical treatment and transanal resection, both modalities fall within the surgical spectrum for rectal cancer management. Transanal resection techniques, such as transanal endoscopic microsurgery (TEM), offer a minimally invasive alternative specifically suited for selected early-stage rectal cancers. However, transanal resection may have limitations in terms of tumor size and lymph node assessment compared to surgical treatment [12].

In the context of endoscopic treatment versus transanal resection, both techniques offer minimally invasive approaches for rectal cancer management. Endoscopic treatment, including TME, is appropriate for selected cases of early-stage rectal cancer, enabling the removal of localized tumors without resorting to open surgery. On the other hand, transanal resection techniques, such as TEM, are better suited for slightly larger tumors or those necessitating full-thickness excision. The choice between endoscopic treatment and transanal resection depends on tumor characteristics, size, and location [12].

In addition, while comparing transanal resection, each has its advantages and considerations. Transanal resection, such as TEM or TAMIS (Transanal Minimally Invasive Surgery), offers the potential for organ preservation and shorter recovery time compared to more extensive surgical procedures [27]. Radiotherapy, particularly when combined with chemotherapy, is often employed in locally advanced cases to improve local control and increase the chances of successful surgical resection [28]. The decision between transanal resection and radiotherapy depends on tumor stage, location, patient preferences, and the expertise available in the healthcare system. Table 1. provides a summary of the main oncological outcomes from studies evaluating surgical, endoscopic, and radiotherapy modalities of treatment. Surgical approaches such as TaTME and TME demonstrated favorable results while endoscopic treatment with TEM had higher recurrence rates. Radiotherapy, particularly combined with chemotherapy, showed efficacy in improving outcomes for rectal cancer.

## 5. Factors Influencing the Choice of Treatment Modality

When considering the choice of treatment modality, several factors come into play. Patient characteristics, such as age, overall health status, and individual preferences, must be taken into account. For instance, younger patients without significant comorbidities may tolerate more invasive surgical interventions, whereas older patients with underlying health issues may benefit from less invasive approaches like endoscopic procedures or transanal resections [39]. 

Tumor characteristics, including stage, location, and histological type, also play a pivotal role in selecting the appropriate treatment modality. Early-stage tumors confined to the mucosa or submucosa may be amenable to endoscopic treatment or transanal resection, while locally advanced tumors often necessitate a multimodal approach involving surgical resection, radiation therapy, and chemotherapy [40]. Multidisciplinary discussions involving surgeons, gastroenterologists, oncologists, and radiologists are paramount to determining the most suitable treatment strategy based on tumor characteristics. Rectal cancer treatments have progressed, yielding improved results for patients. Despite this, there are an array of limitations. One prevalent limitation is the invasiveness of surgical interventions like TME. It also yields potential complications such as urinary or sexual dysfunction or worsened bowel obstruction [25]. In addition, certain individuals cannot have surgery due to health complications or poor overall wellbeing. Another point worth mentioning is how radioactive and chemotherapeutic elements can lead to toxicity affecting various body parts, leading to several side effects. Such effects may include stomach disorders like nausea, vomiting, diarrhea, blisters followed by skin irritations, and blood abnormalities [26].

Healthcare system factors further influence the choice of treatment modality for rectal cancer. The availability of resources, expertise, and multidisciplinary teams can impact the range of treatment options. Specialized centers with experienced surgeons may offer a broader array of surgical techniques and superior outcomes. Access to advanced imaging modalities and radiation oncology services also plays a role in the consideration of radiotherapy. Geographical location and the overall healthcare infrastructure can affect patient access to specific treatments. Collaborative efforts among healthcare providers, regional networks, and referral centers are crucial to ensure that patients receive the most appropriate and optimal treatment considering the available healthcare system factors. Each treatment modality comes with its own advantages, limitations, and applications based on cancer stage (Table 2). Surgical resection is considered the standard but can impact quality of life. Endoscopic techniques are less invasive, except they but have higher recurrence risks. Transanal resection is advantageous for organ preservation, but is inefficient for larger tumors. Radiotherapy improves outcomes for advanced cases. Nevertheless, it carries long-term toxicity risks. Overall, treatment decisions require balancing all constituents, such as tumor factors, patient characteristics, and available resources.

## 6. Discussion

Upon reviewing the literature, it becomes evident that rectal cancer can be effectively treated through three main modalities: surgical intervention, endoscopy therapy, and transanal resection methods such as TEM. Surgical intervention, particularly procedures like total mesorectal excision, is considered the recommended standard due to its high success rate in tumor removal, favorable long-term survival outcomes, and low rates of local recurrence [7,69]. TME is the recommended surgical procedure, which involves extracting the rectal tumor and pararectal lymph nodes for better pathological assessment while preserving the surrounding tissues outside the rectal fascia [7,8].

The surgical options have shown promising results with low rates of local recurrence and improved survival rates. The selection of patients for surgical treatment takes into account factors such as the stage of the disease, overall health, and individual preferences [70]. The outcomes of surgical treatment are evaluated based on survival rates, local recurrence rates, distant metastasis rates, postoperative complications, and quality of life measures [70]. The recovery period following surgery can be lengthy, requiring a significant commitment of time. No improvement in 3-year disease-free survival was observed for TaTME over laparoscopic TME for rectal cancer patients [35]. Equivalent outcomes were demonstrated between TME and transabdominal TME with regards to survival, recurrence, and disease-free survival based on the meta-analysis [71].

Endoscopic treatment, on the other hand, plays an significant role as it represents a less invasive alternative, especially for early-stage rectal tumors. It is typically used in patients with superficial lesions or those who are ineligible for surgery due to comorbidities [15]. Endoscopic procedures are used to remove or ablate the tumor, thereby avoiding complex surgical interventions. This approach encompasses techniques such as TME, albeit with a higher risk of local recurrence compared to surgical treatment. While endoscopic treatment can be effective in selected cases, its utility may be limited when dealing with more advanced or complex tumors [69].

Endoscopic treatments, including TME, although effective for early-stage rectal cancer, carry higher risks compared to surgical interventions. Transanal methods like TEM offer a minimally invasive and efficient treatment alternative for certain early-stage rectal cancers, but limitations exist in terms of lymph node assessment and suitability for larger tumors. TEM is the gold standard endoscopic procedure for precisely removing the rectal tumor and surrounding mesorectal tissue [69]. TEM aims to minimize the risk of local recurrence and improve long-term outcomes. Different endoscopic procedures are employed based on the characteristics of the tumor. Polypectomy and local excision are used for small tumors or polyps confined to the inner layers of the rectal wall [72]. Endoscopic mucosal resection (EMR) is performed for larger, superficial tumors located in the mucosal layer of the rectum [40]. Endoscopic submucosal dissection (ESD) allows for block resection of larger, non-lifting tumors [40]. Patient selection for endoscopic treatment considers factors such as tumor size, location, stage, and absence of lymph node involvement or distant metastasis [40]. The outcomes of endoscopic treatment vary based on tumor features, patient selection, and the extent of the resection. Preservation of anal function and quality of life measures are important considerations.

Endoscopic treatment offers advantages such as its less invasive nature compared to surgery, shorter recovery time, and potential for preserving anal function [73]. However, it carries limitations such as the risk of incomplete tumor removal, which may necessitate additional treatments. Careful patient selection, consideration of tumor characteristics, and the availability of experienced endoscopists are crucial for achieving optimal outcomes.

Transanal resection is another technique used for the removal of rectal tumors. The rectum is accessed through the anus and the resection is performed with special instruments. It allows local tumor removal without the need for extensive abdominal surgery [73]. Transanal resection can be performed as an individual surgical procedure or in combination with other surgical approaches. The technique of transanal resection varies depending on the size and location of the tumor. It may involve local excision or more extensive resections such as transanal endoscopic microsurgery (TEM), which allows for precise tumor removal while preserving anal function [12]. Patient selection for transanal resection considers factors such as tumor stage, size, location, absence of lymph node involvement or distant metastasis, overall health, preferences, and the expertise and resources available. The outcomes of transanal resection depend on tumor features, patient selection, and the extent of the resection. Preservation of anal function and quality of life measures are important considerations.

Transanal resection offers advantages such as its minimally invasive nature, potential for preserving anal function, and shorter recovery time compared to more extensive surgical procedures [12]. However, it carries a risk of local recurrence and may not be feasible or appropriate for all cases. Careful patient selection and consideration of tumor characteristics are essential for achieving optimal outcomes.

Radiotherapy also plays a vital role in managing rectal cancer by acting as a primary treatment and a supplementary measure alongside surgery. This technique employs powerful radiation to eliminate cancer cells or hinder their growth [74]. There is a range of radiotherapy modalities available [75]. EBRT (external beam radiation therapy), the predominant method, involves directing radiation from an external source, typically using a linear accelerator, to the tumor site. It can be administered before surgery (neoadjuvant), after surgery (adjuvant), or as the palliative treatment for unresectable or metastatic disease [75]. Neoadjuvant radiotherapy refers to delivering radiation before surgery to reduce tumor size and enhance surgical outcomes. In contrast, brachytherapy entails placing radioactive sources directly into or near the tumor, serving as a boost therapy in combination with EBRT or as the primary treatment in specific cases [75]. Neoadjuvant chemoradiotherapy has become an integral part of the treatment approach for locally advanced rectal cancer. It typically involves a combination of chemotherapy and radiation therapy [76]. Adjuvant therapy, administered after surgery, may be recommended for patients with high-risk features, such as incomplete tumor removal [77]. 

The impact of radiotherapy on rectal cancer treatment outcomes is influenced by a variety of factors. These include the chosen treatment approach, the stage of the tumor, and the careful selection of patients. When used in conjunction with chemotherapy, radiotherapy has demonstrated positive outcomes by enhancing local control and minimizing the likelihood of local recurrence [78]. It has also been linked to improved overall survival and disease-free survival, particularly in cases of locally advanced cancer [28]. Advances in radiotherapy techniques, such as IMRT (intensity-modulated radiation therapy), have allowed for more precise targeting of tumor tissues while sparing healthy surrounding tissues. This can lead to improved efficacy and reduced toxicity. In addition, the development of techniques like SBRT (stereotactic body radiation therapy) has expanded the treatment options and may provide benefits in select cases. By individualizing treatment plans and considering the specific characteristics of each patient’s tumor and overall health, healthcare providers can optimize the outcomes and minimize the limitations associated with the current standard of care. Nevertheless, it is crucial to consider the potential adverse effects linked to radiotherapy. These encompass long-lasting modifications in the functioning of the bowels or urinary system that persist over a prolonged duration [79].

Immunotherapy is a promising treatment option for patients with colo-rectal cancer, particularly those with MSI-H/dMMR status [80,81]. MSI-H/dMMR colorectal cancer (CRC) accounts for 12–15% of all CRC cases and is more likely to occur in right-sided cancers. However, only 2% of patients with rectal cancer are MSI-H. Furthermore, a retrospective review of 73 patients with confirmed dMMR/MSI-H colorectal cancer who received any type of programmed cell death-1 (PD-1) inhibitor prior to surgery showed that 84.9% of patients achieved an objective response, with 23.3% showing a complete response, and 61.6% a partial response. The 2-year overall and disease-free survival rates were 100% for patients who underwent surgery after being treated with PD-1 inhibitors [82]. In the KEYNOTE-177 trial, pembrolizumab showed superior progression-free survival compared to chemotherapy in MSI-H/dMMR stage IV colorectal carcinoma [83].

The findings from the literature review align with existing knowledge regarding rectal cancer treatment and can inform health policy decisions regarding rectal cancer treatment. Surgical interventions, such as TME, are widely accepted due to their demonstrated success in achieving better oncological outcomes. These procedures ensure proper tumor removal, resulting in lower rates of local recurrence and improved long-term survival. Clinicians should prioritize surgical treatment, particularly TME, as the standard approach for rectal cancer whenever possible. While endoscopic therapies and transanal resection techniques show potential benefits for early-stage tumors, they are not sustainable for advanced cases but the risk of local recurrence should be carefully considered. Surgeons should acquire expertise in TME and transanal resection techniques, and healthcare centers should have access to multidisciplinary teams and advanced imaging modalities for accurate staging and treatment planning.

Implementing these findings in clinical practice may require additional training and resources. This will help to enhance the quality of care provided to rectal cancer patients and improve treatment outcomes. Health policymakers should recognize the importance of ensuring access to specialized centers with experienced surgeons and multidisciplinary teams.

Consequently, the review revealed gaps in our current knowledge of rectal cancer therapy, like the need for additional research comparing the relative effectiveness and long-term effects of various treatments, as well as a deeper investigation of how patient traits, tumor characteristics, and healthcare system aspects influence the choice of therapy and impact results. Additionally, there remains a great need for research into cutting-edge approaches, like targeted therapies and minimally invasive surgical procedures, to fill in these gaps and improve the treatment choices for rectal cancer. Further studies directly comparing the relative efficacy and long-term outcomes of different treatment approaches are needed. Research focusing on how patient characteristics, tumor features, and healthcare system factors influence treatment decisions and impact outcomes is essential. Additionally, evaluating innovative methods such as minimally invasive surgery techniques and targeted therapies can contribute to filling these knowledge gaps and improving rectal cancer treatment.

## 7. Conclusions

In conclusion, surgical treatment, particularly TME, remains the recommended standard for rectal cancer due to its proven efficacy in achieving complete tumor resection and improving long-term outcomes. Endoscopic treatment and transanal resection techniques offer minimally invasive alternatives for selected early-stage cases. Depending on the specific characteristics of the patient and the tumor, treatment choices should be made individually while taking into account the system’s resources and knowledge. To further enhance the results of individualized therapy in rectal cancer cases, future research should concentrate on filling in the information gaps that have been discovered. 

## Figures and Tables

**Table 1 healthcare-11-02347-t001:** Oncological outcomes of different rectal cancer treatment procedures.

Procedure	No. of Patients	Local Recurrence Rate	Disease-Free Survival (DFS)	Overall Survival (OS)
TaTME [29]	159	3-year: 2.0%, 5-year: 4.0%	3-year: 92%, 5-year: 81%	3-year: 83.6%, 5-year: 77.3%
TaTME [30]	2803	2-year: 4.8%	2-year: 77%	2-year: 92%
TEM [31]	100	5-year: 12.8%, 10-year: 15.6%	-	-
TaTME [32]	624	2-year: 4.6%, 3-year: 6.6%	-	-
Radiotherapy (Comparative study) [33]	154 patients were randomly assigned to short (n = 77) or long interval (n = 77)	Short-course: 1.3% Long-course: 11.7% (*p* = 0.03)		Short-course: 10-year survival—58% Long-course: 10-year survival—61% (*p* = 0.754)
Radiotherapy [34]	271	4.40%	5-year: 65.5%, 10-year: 51%	5-year: 73%, 10-year: 55.5%
TaTME, LapTME (Comparative study) [35]	126 (TaTME), 126 (LapTME)	TaTME: 9.5%, LapTME: 23.8%	TaTME: 3-year: 80.3%, LapTME: 3-year: 73.6%	TaTME: 3-year: 92%, LapTME: 3-year: 92.9%
TaTME [36]	-	1-year: 94.8%, 2-year: 89.3%, 3-year: 80.2%	1-year: 97.4%, 2-year: 95.7%, 3-year: 92.9%	
TaTME, LapTME (Comparative study) [37]	25 (TaTME), 38 (LapTME)	TaTME: 9.5%, LapTME: 23.8%	TaTME: 3-year: 74.3%	TaTME: 3-year: 90.9%, LapTME: 84.2%
TME [38]	-	2-year: 3.6%	2-year: 91% 3-year: 88% 4-year: 85%	-

**Table 2 healthcare-11-02347-t002:** Rectal cancer treatment modalities: a literature matrix synthesis.

Procedure	Efficacy	Side Effects	Applicability (Stages)	Quality of Life Impact
TEM (total endoscopic microsurgery)	Effective for early-stage tumors and rectal adenomas, with good local control and low recurrence rates [41,42,43].	Low rates of clinically relevant postoperative morbidity [44].	Most effective for early-stage rectal cancer, particularly Tis and T1 N0 M0 tumors [45,46]. Used for higher-stage cancers after neoadjuvant therapy has been explored [47].	Reduced surgical trauma, fewer complications, and rapid postoperative recovery [48].
taTME (transanal total mesorectal excision)	Offers better visualization of the distal rectum and ability to perform deep pelvic dissection [49,50]. Lower conversion rate compared to laparoscopic TME [9].	Associated with a significant rate of intraoperative complications during the learning curve [51].	Primarily indicated for mid- and low-rectal cancer.	Initially decreases quality of life scores, but most scores return to baseline values after 6 months [52].
TME (total mesorectal excision)	High efficacy in providing improved local control and survival in early-stage and locally advanced rectal cancer (LARC) [53].	Significant morbidity, bowel dysfunction, sexual and urinary dysfunction [54,55,56].	Most effective in LARC [54].	Major long-term side effects include major low-anterior resection syndrome (LARS), which can significantly impact the quality of life [57].
Radiotherapy	Effective as a palliative treatment for symptomatic rectal cancer [58]. Improves outcomes when combined with chemotherapy for locally advanced rectal cancer [59,60,61].	Anal and urinary dysfunction, cardiovascular morbidity, and radiation proctopathy [62,63].	Standard of care for locally advanced rectal cancer. Short-term preoperative radiotherapy reduces toxicity and prevents local relapse [64,65]. Adjuvant radiotherapy improves cause-specific survival in certain stages [66].	Long-term effects on anorectal function, leading to bowel frequency, fecal incontinence, and other functional problems [67,68].

LARC—locally advanced rectal cancer.

## Data Availability

Not applicable.

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
