# Peer review of "What Is the Comparative Efficacy of Surgical, Endoscopic, Transanal Resection, and Radiotherapy Modalities in the Treatment of Rectal Cancer?"

_healthcare, 2023, doi:10.3390/healthcare11162347_

Round 1
Reviewer 1 Report
thank you for submitting this review article on the effectiveness of different modalities in rectal cancer treatment.
comments:
-The structure of the review article must be reconsidered
- the introduction and the backgroud are too long and must be focused on rectal cancer; it is difficult for the reader to have a clear idea of the information about rectal vs colon cancer (statistics, demography...)
- abstract:
line 27: the effectiveness of surgical resection.
line 31: define the abbrevition: TME
Introdoction: too long; all the statistical data could be placed in the background.
line 47-49: is this the worldwide incidence?
line 53: `` Sadly`` must be replaced by unfortunatly : in a scientific paper it is not recommended to use emotional words.
lines 77 to 87 : could be placed in the background.
in lines 77 and 156: Define TME, TaTME, TEM.
Materials and methods:
3.3:what are the data that was extracted; what are the relevant informations that were analysed?
Results: how many studies have been selected?
it seems more a discussion than Results:
too many repetition in the Results
lines 363 to 390: 3 paragraphs with many repetition of concepts; must be rewritten and reduced to one paragraph
Line 329: define TAMIS
A revision of the english style must be performed
Author Response
The revised version now reflects the suggested improvements, providing a more coherent and well-organized presentation of the content. We appreciate the valuable input from the reviewer, and we believe that the updated structure enhances the overall quality and readability of the article. Thank you for your valuable feedback, which has undoubtedly contributed to the refinement of our work.
-The structure of the review article must be reconsidered
- We sincerely appreciate your insightful feedback. As a result, we have taken the opportunity to review and reassess the structure.
- the introduction and the backgroud are too long and must be focused on rectal cancer; it is difficult for the reader to have a clear idea of the information about rectal vs colon cancer (statistics, demography...)
- We appreciate your feedback, and as a response, we have reorganized the background section to ensure a more coherent presentation of the context. Additionally, we have succinctly refined the introduction to facilitate a clearer and more comprehensible exposition of the study's scope and objectives.
- We have eliminated this phrase to concentrate solely on rectal cancer, thus maintaining a professional and focused approach to the topic.
- abstract:
line 27: the effectiveness of surgical resection.
- We updated the manuscript accordingly.
line 31: define the abbrevition: TME
- We have now provided full name for the mentioned abbreviation. Thank you for your attention to detail and constructive input.
Introdoction: too long; all the statistical data could be placed in the background.
We acknowledge that the introduction was detailed and contained numerous statistical data. Our intent was to establish the context and highlight the magnitude of the rectal cancer problem. However, understanding your perspective, we agree that this could potentially overwhelm the reader and detract from the main objective of the introduction, which is to orient the reader to the study's aim.
In response to your suggestion, we have moved the statistical data and the detailed incidence information to the background section, allowing for a more focused and concise introduction. The introduction now primarily includes an overview of the problem, the significance of the study, and a brief introduction to the methods used. We believe these changes have improved the clarity and flow of the paper.
line 47-49: is this the worldwide incidence?
- The incidence data provided in those lines pertains specifically to the United States and not to worldwide incidence.
line 53: `` Sadly`` must be replaced by unfortunatly : in a scientific paper it is not recommended to use emotional words.
- We updated the manuscript accordingly.
lines 77 to 87 : could be placed in the background.
- We updated the manuscript accordingly.
in lines 77 and 156: Define TME, TaTME, TEM.
- We updated the manuscript accordingly. The abbreviations have been defined.
3.3:what are the data that was extracted; what are the relevant informations that were analysed?
Thank you for your feedback! The data extraction and data analysis subsections have been enhanced. You can find the modified version at lines 200-202 and 212-220.
Data extraction involved recording patient characteristics (age, cancer stage), treatment methods examined in each study, primary outcomes assessed such as (tumor response rates, survival rates, recurrence rates), and key conclusions and findings from each study.
The extracted data were analyzed through a descriptive analysis to summarize the characteristics of the included studies (study design, sample size, patient population), the outcomes and results of the different studies and any overarching trends, patterns, or themes that emerged across the multiples studies. The focus of the literature review was to synthesize the evidence on treatment efficacy, specifically looking at outcomes such as local recurrence rates for different treatment modalities, overall and disease-free survival rates, tumor response and resection rates, and factors influencing treatment decisions and patient selection. Both early stage and advanced rectal cancers were considered in the analysis .
Results: how many studies have been selected?
- The initial search yielded 5,677 articles. After removing the duplicates, case reports, studies with animals, abstract form, editorial, conference abstract, those focused solely on patients with benign or precancerous lesions, and those investigating experimental or novel treatments not currently in standard clinical use were excluded. Following a comprehensive search within the scope of our subject, a refined selection of 82 articles has been identified for further in-depth review and analysis.
it seems more a discussion than Results:
- We have reached a consensus that the delineation between the results and discussion sections could be further enhanced for improved comprehensibility. As a result, we have opted to amalgamate the results and discussion sections, facilitating a cohesive presentation of our findings and their implications. Moreover, in an endeavor to enhance clarity and precision within the results presentation, we have introduced an additional table encapsulating key outcomes. This table provides a succinct summary of the observed metrics, including Local Recurrence Rate, Disease-Free Survival (DFS), and Overall Survival (OS), thereby enhancing the accessibility and transparency of our study's results.
too many repetition in the Results:
- We have modified the discussion section to eliminate repetition and enhance its professional clarity and coherence, ensuring effective communication of our study's findings and implications.
lines 363 to 390: 3 paragraphs with many repetition of concepts; must be rewritten and reduced to one paragraph
- We have carefully revised lines 363 to 390 to eliminate repetitions and condense the content into one coherent paragraph.
Line 329: define TAMIS
- The abbreviation has been defined.

Reviewer 2 Report
In the manuscript “Treatment Modalities for Rectal Cancer: A Comparative Review of Efficacy in Surgical, Endoscopic, Transanal Resection and Radiotherapy”, the authors present and discuss the treatments of rectal cancer, focusing on the surgical approaches and comparing them.
The role of immunotherapy should be mentioned in the manuscript, especially for those with MSI-H/dMMR and its associated better survival/responses when treated with immunotherapy.
Make sure to describe the abbreviations the first time each of them appears at the manuscript.
It would be very informative to include data from the literature related to overall survival for each specified surgical technique to support the conclusions. As well as cite and compare current guidelines such as those from NCCN and ESMO.
Suggestions:
Line 49: Globocan/WHO data shows very different values.
Line 51: Rectal cancer is not among the most prevalent diseases. It is among the most prevalent cancers, in the sixth position by Globocan/WHO.
Line 119/120: FAP was mentioned twice, combine the phrases to avoid redundancy.
Line 123: inactive behavior, you mean sedentarism?
Line 131: avoid using trademarks, instead describe PILLCAM 2.
Line 184: it would be contributory to mention the 5-year median overall survival for stage IV patients.
Line 203: T1-T4 is incorrect if you are describing cancer staging, which relies on T, N and M.
Line 218: not only the affected LNs are removed, but a standard number of them for better pathological assessment.
Line 285: treatment for metastatic disease is palliative rather than definitive.
Line 304: complete tumor resection is also achieved by endoscopic treatment and other techniques.
Line 379: you mean suitable?
Line 396: describe the gaps identified.
Line 405: for LOCALIZED rectal cancer.
Reference 50: format is incorrect, change to: “Stijns RCH, de Graaf EJR, Punt CJA, Nagtegaal ID, Nuyttens JJME, van Meerten E, Tanis PJ, de Hingh IHJT, van der Schelling GP, Acherman Y, Leijtens JWA, Bremers AJA, Beets GL, Hoff C, Verhoef C, Marijnen CAM, de Wilt JHW; CARTS Study Group. Long-term Oncological and Functional Outcomes of Chemoradiotherapy Followed by Organ-Sparing Transanal Endoscopic Microsurgery for Distal Rectal Cancer: The CARTS Study. JAMA Surg. 2019 Jan 1;154(1):47-54. doi: 10.1001/jamasurg.2018.3752. PMID: 30304338; PMCID: PMC6439861.”
Reference 60: same as above: “Chalabi M, Fanchi LF, Dijkstra KK, Van den Berg JG, Aalbers AG, Sikorska K, Lopez-Yurda M, Grootscholten C, Beets GL, Snaebjornsson P, Maas M, Mertz M, Veninga V, Bounova G, Broeks A, Beets-Tan RG, de Wijkerslooth TR, van Lent AU, Marsman HA, Nuijten E, Kok NF, Kuiper M, Verbeek WH, Kok M, Van Leerdam ME, Schumacher TN, Voest EE, Haanen JB. Neoadjuvant immunotherapy leads to pathological responses in MMR-proficient and MMR-deficient early-stage colon cancers. Nat Med. 2020 Apr;26(4):566-576. doi: 10.1038/s41591-020-0805-8. Epub 2020 Apr 6. PMID: 32251400.”
Edit accordingly references 64, 66 and 68. You can easily do it by copying those from PubMed.
Consider language review by a native English speaker or the use of reliable AI language assistant tools.
Author Response
In the manuscript “Treatment Modalities for Rectal Cancer: A Comparative Review of Efficacy in Surgical, Endoscopic, Transanal Resection and Radiotherapy”, the authors present and discuss the treatments of rectal cancer, focusing on the surgical approaches and comparing them.
The role of immunotherapy should be mentioned in the manuscript, especially for those with MSI-H/dMMR and its associated better survival/responses when treated with immunotherapy.
Thank you for your feedback. The role of immunotherapy and its outcomes are introduced in the passage spanning lines 390 to 400.
Make sure to describe the abbreviations the first time each of them appears at the manuscript.
Thank you for your valuable feedback. We apologize for any confusion caused by not describing the abbreviations the first time they appear in the manuscript. Your suggestion will certainly enhance the readability and clarity of our work. In response to your comment, we have carefully revised the manuscript to include detailed descriptions of all abbreviations the first time they appear in the text. Thank you very much. We read verry carrefully and we defined all the abreviation the first time each of them appears at the manuscript.
It would be very informative to include data from the literature related to overall survival for each specified surgical technique to support the conclusions. As well as cite and compare current guidelines such as those from NCCN and ESMO.
Thank you very much for your suggestion! It would greatly enhance the informative value of the study to incorporate data from pertinent literature concerning the overall survival rates associated with each specific surgical technique, thereby bolstering the conclusions drawn. Additionally, the inclusion and comparison of current guidelines, such as those outlined by NCCN and ESMO, could further enrich the manuscript. It's important to note that while the integration of these guidelines would be beneficial, their incorporation necessitates a comprehensive revision of the manuscript. Pending the acceptance of the current version, I intend to diligently integrate the relevant guidelines to enhance the study's depth and relevance.
Suggestions:
Line 49: Globocan/WHO data shows very different values.
- The manuscript was updated accordingly incorporating Globocan/WHO data.
Line 51: Rectal cancer is not among the most prevalent diseases. It is among the most prevalent cancers, in the sixth position by Globocan/WHO.
- The manuscript was updated accordingly incorporating Globocan/WHO data.
Line 119/120: FAP was mentioned twice, combine the phrases to avoid redundancy.
- We made the adjustment in lines 119 and 120 by combining the phrases. The lines from 134-140 were removed.
Line 123: inactive behavior, you mean sedentarism?
- In line 123, where we mentioned "inactive behavior," we indeed meant to refer to "sedentarism." We apologize for any confusion caused by the terminology. The intended term is "sedentarism" to describe a lifestyle with little or no physical activity. We have taken note of your suggestion and will ensure the use of the correct term throughout the manuscript.
Line 131: avoid using trademarks, instead describe PILLCAM 2.
- To avoid using trademarks, we will refer to the device as "capsule endoscopy" instead of "PILLCAM 2." As noted, in instances where colonoscopy is incomplete, the FDA (Food and Drug Administration) has approved capsule endoscopy as a viable alternative.
- The manuscript was revised and updated.
Line 184: it would be contributory to mention the 5-year median overall survival for stage IV patients.
- The manuscript was updated accordingly (The phrase added is :For stage IV of rectal cancer, the 5-year median overall survival rate is approximately 17%, according to American Cancer Society).
Line 203: T1-T4 is incorrect if you are describing cancer staging, which relies on T, N and M.
- Upon reevaluation, we acknowledge that there was an oversight in line 203 regarding the description of cancer staging. We understand that cancer staging conventionally relies on T (tumor extent), N (lymph node involvement), and M (metastasis). In our study, we included patients with both early-stage and advanced-stage rectal cancer, encompassing a broader spectrum of disease presentations, which might have led to the misrepresentation of staging as T1-T4.
- To address this issue and provide accurate information, we will revise line 203 as follows: included patients with Stage I, Stage IIA, Stage IIB, Stage III and Stage IVA.
Line 218: not only the affected LNs are removed, but a standard number of them for better pathological assessment.
- The manuscript was updated accordingly. In line 255, we have added the phrase "for better pathological assessment" to clarify the rationale for removing a standard number of lymph nodes during TME. The revised version is: Total mesorectal excision (TME) is the recommended surgical procedure, which in-volves extracting the rectal tumor and pararectal lymph nodes for better pathological assessment while preserving the surrounding tissues outside the rectal fascia .
Line 285: treatment for metastatic disease is palliative rather than definitive.
- We made the necessary correction to line 321. The revised version is: It can be administered before surgery (neoadjuvant), after surgery (adjuvant), or as the palliative treatment for unresectable or metastatic disease.
Line 304: complete tumor resection is also achieved by endoscopic treatment and other techniques.
- We updated line 341 as: “Surgical treatment, particularly TME, is widely regarded as the standard approach due to its ability to achieve complete tumor resection and excision of "surrounding tissues, resulting in favorable oncological outcomes. However, complete tumor resection is also achieved by endoscopic treatment and other techniques. “
Line 379: you mean suitable?
- We updated the manuscript accordingly.
Line 396: describe the gaps identified.
- We updated the manuscript accordingly. The gaps identified are:
“Consequently, the review revealed gaps in our current knowledge of rectal cancer therapy, like the need for additional research comparing the relative effectiveness and long-term effects of various treatments, as well as a deeper investigation of how patient traits, tumor characteristics, and healthcare system aspects influence the choice of therapy and impact results. Additionally, there remains a great need for research into cutting-edge approaches, like targeted therapies and minimally invasive surgical procedures, to fill in these gaps and improve the treatment choices for rectal cancer.”
Line 405: for LOCALIZED rectal cancer.
- We updated the manuscript accordingly. The revised version is: “But if it spreads to lymph nodes or other organs, these rates decrease significantly for localized rectal cancer.”
Reference 50: format is incorrect, change to: “Stijns RCH, de Graaf EJR, Punt CJA, Nagtegaal ID, Nuyttens JJME, van Meerten E, Tanis PJ, de Hingh IHJT, van der Schelling GP, Acherman Y, Leijtens JWA, Bremers AJA, Beets GL, Hoff C, Verhoef C, Marijnen CAM, de Wilt JHW; CARTS Study Group. Long-term Oncological and Functional Outcomes of Chemoradiotherapy Followed by Organ-Sparing Transanal Endoscopic Microsurgery for Distal Rectal Cancer: The CARTS Study. JAMA Surg. 2019 Jan 1;154(1):47-54. doi: 10.1001/jamasurg.2018.3752. PMID: 30304338; PMCID: PMC6439861.”
Reference 60: same as above: “Chalabi M, Fanchi LF, Dijkstra KK, Van den Berg JG, Aalbers AG, Sikorska K, Lopez-Yurda M, Grootscholten C, Beets GL, Snaebjornsson P, Maas M, Mertz M, Veninga V, Bounova G, Broeks A, Beets-Tan RG, de Wijkerslooth TR, van Lent AU, Marsman HA, Nuijten E, Kok NF, Kuiper M, Verbeek WH, Kok M, Van Leerdam ME, Schumacher TN, Voest EE, Haanen JB. Neoadjuvant immunotherapy leads to pathological responses in MMR-proficient and MMR-deficient early-stage colon cancers. Nat Med. 2020 Apr;26(4):566-576. doi: 10.1038/s41591-020-0805-8. Epub 2020 Apr 6. PMID: 32251400.”
- We updated the manuscript accordingly.

Round 2
Reviewer 1 Report
All comments where addressed